# Emerging *Clostridioides difficile* ribotypes have divergent metabolic phenotypes

Firas S. Midani,[1,2] Heather A. Danhof,[1,2] Nathanael Mathew,[1,2] Colleen K. Ardis,[1,2] Kevin W. Garey,[3] Jennifer K. Spinler,[4] Robert A. Britton[1,2]

**ABSTRACT** *Clostridioides difficile* is a gram-positive spore-forming pathogen that commonly causes diarrheal infections in the developed world. Although *C. difficile* is a genetically diverse species, certain ribotypes are overrepresented in human infections, and it is unclear if metabolic adaptations are essential for the emergence of these epidemic ribotypes. To identify ribotype-specific metabolic differences, we therefore tested carbon substrate utilization by 88 *C. difficile* isolates and looked for differences in growth between 22 ribotypes. As expected, *C. difficile* was capable of growing on a variety of carbon substrates. Further, *C. difficile* strains clustered by phylogenetic relationship and displayed ribotype-specific and clade-specific metabolic capabilities. Surprisingly, we observed that two emerging lineages, ribotypes 023 and 255, have divergent metabolic phenotypes. In addition, although *C. difficile* Clade 5 is the most evolutionary distant clade and often detected in animals, it displayed robust growth on simple sugars similar to Clades 1–4. Altogether, our results corroborate the generalist metabolic strategy of *C. difficile* but also demonstrate lineage-specific metabolic capabilities.

**IMPORTANCE** The gut pathogen *Clostridioides difficile* utilizes a wide range of carbon sources. Microbial communities can be rationally designed to combat *C. difficile* by depleting its preferred nutrients in the gut. However, *C. difficile* is genetically diverse with hundreds of identified ribotypes, and most of its metabolic studies were performed with lab-adapted strains. To identify ribotype-specific metabolic differences, we profiled carbon metabolism by a myriad of *C. difficile* clinical isolates. While the metabolic capabilities of these isolates clustered by their genetic lineage, we observed surprising metabolic divergence between two emerging lineages. We also found that genetically newer and older clades grew to a similar level on simple sugars, which contrasts with recent findings that newer clades experienced positive selection on genes involved in simple sugar metabolism. Altogether, our results underscore the importance of considering the metabolic diversity of pathogens in the study of their evolution and the rational design of therapeutic interventions.

**KEYWORDS** *Clostridioides difficile*, growth modeling, carbon metabolism, ribotyping

Clostridioides difficile is a common cause of gastrointestinal infections with symptoms ranging from mild diarrhea to pseudomembranous colitis, which can be fatal (1). By disrupting the gut microbiome, antibiotic use lessens microbial competition for nutrients in the gut and increases the risk of *C. difficile* infection (2). Although hospital-acquired *C. difficile* infections are decreasing likely due to better infection-prevention and antibiotic stewardship efforts, community-acquired *C. difficile* infections are increasing (3). Therefore, nutritional and microbial alterations in the gut due to factors other than antibiotics may also increase the risk of *C. difficile* infections. As a strong driver of both

**Peer Reviewers** Jordy Evan Sulaiman, The Hong Kong University of Science and Technology, Hong Kong, China; Joseph P. Zackular, University of Pennsylvania, Philadelphia, Pennsylvania, USA

Address correspondence to Firas S. Midani, firas.midani@bcm.edu, or Robert A. Britton, robert.britton@bcm.edu.

The authors declare no conflict of interest.

See the funding table on p. 15.

nutritional and microbial composition in the gut (4), the human diet modulates the risk or severity of *C. difficile* infection in animals (5, 6), and dietary additives may have contributed to the emergence of epidemic *C. difficile* lineages (7). A deeper understanding of the nutritional preferences of *C. difficile* can therefore inform dietary and microbial strategies for the prevention and treatment of *C. difficile* infections.

As a generalist species, *C. difficile* occupies various nutritional niches in the human gut. Because it is adept at fermenting both amino acids and carbohydrates (8), *C. difficile* thrives during antibiotic disturbance (2, 9), chronic inflammation (10), or toxin-induced inflammation (11, 12). However, this rich understanding of *C. difficile* carbon metabolism was derived from animal studies using a few lab-adapted *C. difficile* strains, and confounded by the phenotypic divergence of the same *C. difficile* strain passaged in different laboratories (13). Additionally, *C. difficile* is a diverse species that has five well-defined phylogenetic clades (14) and at least 116 PCR ribotypes (15). Metabolic capabilities, including fermentation of amino acids and central carbon metabolites, also varied considerably between strains comprising the five major clades (16). Yet, how these capabilities vary between different ribotypes remains incompletely understood. Broader studies of *C. difficile* metabolism are therefore needed to identify ribotype-specific metabolic capabilities. By discovering metabolic differences between dominant, emerging, and rare ribotypes, these studies can also improve our understanding of *C. difficile* evolution and epidemiology.

Accordingly, we measured and compared the growth of 88 *C. difficile* isolates on 190 carbon substrates. By profiling growth in Biolog Phenotype MicroArray (PM) plates, we identified a wide range of carbon sources used by the *C. difficile* species and compared how ribotypes differentially grew on these substrates. We also contrasted the growth of the emerging ribotypes 023 and 255 as well as the growth of ancestral Clade 5 ribotypes to newer Clades 1–4 ribotypes. Altogether, our results expand our understanding of the metabolic diversity of *C. difficile* and pose new questions about the evolution of the *C. difficile* species.

## RESULTS

### Broad survey of carbon substrate utilization by various *C. difficile* ribotypes

We profiled the growth of 88 *C. difficile* strains on 190 unique carbon substrates using Biolog Phenotype MicroArray plates. Tested strains comprised all five phylogenetic clades and at least 22 ribotypes. Strains profiled were from patient samples ($n = 82$) or lab-adapted isolates ($n = 6$), including CD630, R20291, VPI 10463, and M68. Strains were grown anaerobically for 17 h in Biolog plates which are pre-loaded with a single carbon substrate in each well. Under certain media conditions, *C. difficile* cultures experience abrupt declines in optical density during the stationary phase (17, 18). So, we estimated the maximum population size for each monoculture using the carrying capacity metric (total growth), which is defined as the difference between the initial and maximum optical density (OD) during the growth experiments. Because *C. difficile* grows in minimal media using Stickland fermentation of amino acids, maximum changes in optical density reflect the additive growth of amino acids in the minimal media and on the single carbon source pre-loaded in each well. We therefore adjusted the carrying capacities for each isolate by subtracting its carrying capacity on minimal media alone in the control well (normalized carrying capacity). This normalization allowed us to discriminate the contribution of a carbon source versus amino acids to bacterial growth and account for variation in the growth of different isolates on minimal media.

### *C. difficile* is a generalist species that grows on a variety of carbon substrates

*C. difficile* isolates grew on a variety of carbon substrates. Out of 190 tested carbon sources, 26 substrates increased optical density by at least 0.1 units in at least 10% of the tested isolates (Fig. 1). To rank these substrates based on their contribution to *C. difficile* growth, we ordered them by their median normalized carrying capacities. *C.*

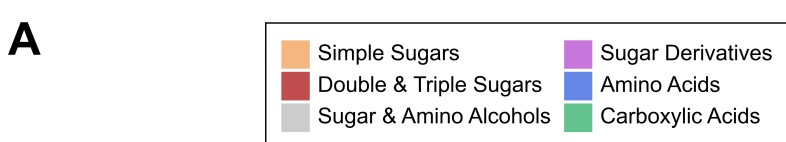

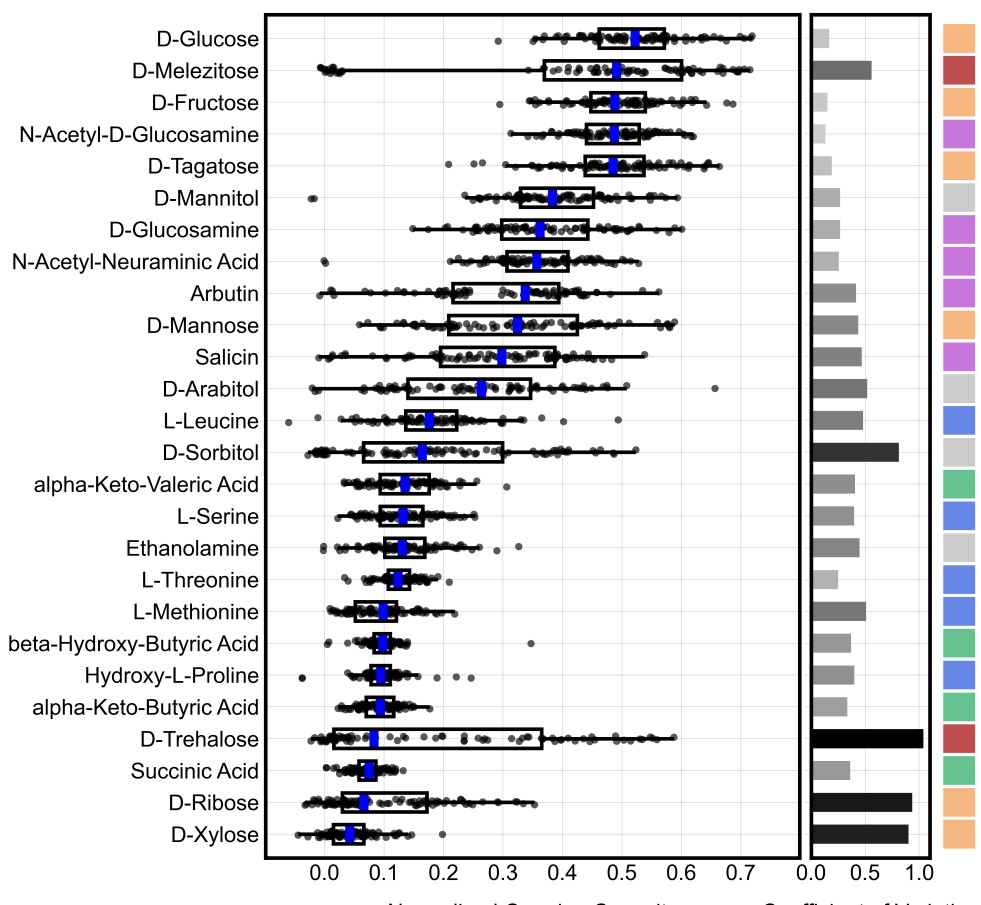

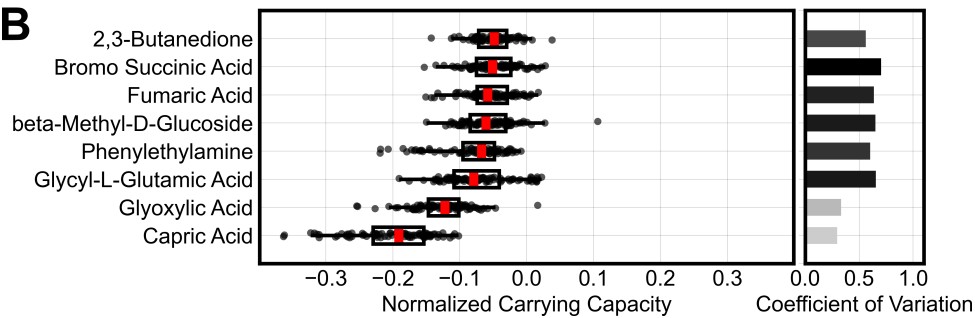

**FIG 1** *C. difficile* is a generalist species that grows on a variety of carbon sources. (A) Plot displays the normalized carrying capacity for 88 *C. difficile* isolates on the top 26 most commonly used substrates. Normalized carrying capacity is defined as the optical density on minimal media supplemented with a single carbon source beyond the optical density on minimal media alone. (B) Bottom panel displays carbon sources that inhibited growth. Box plots display the median and interquartile range of values. Whiskers extend to the farthest point within 1.5× of the interquartile range. Horizontal bars on the right side display the coefficients of variation for each distribution, and the shading of these bars scales with the coefficient's value. For inhibitory substrates, coefficients of variation were computed on −1× normalized carrying capacity. Top legend maps each substrate to a chemical group.

*difficile* reached the highest biomass on simple sugars, followed by sugar derivatives, sugar alcohols, amino acids, and then carboxylic acids. However, because baseline amino acids in the minimal media are fermented by *C. difficile*, normalized carrying capacities for amino acids underestimate their total contribution to *C. difficile* growth. Conversely, our survey also revealed several substrates that decreased optical density by a mean of at least 0.05 units. These inhibitory substrates were dominated by acids and included substrates with known antimicrobial effects, such as capric acid, and those that can be microbially produced, such as phenylethylamine (19, 20). For growth on the top 26 substrates, we also estimated growth rates for all isolates (Fig. S1) and found that medians of normalized carrying capacities were highly correlated with medians of normalized growth rates (Pearson's correlation coefficient = 0.87, $P < 0.001$). However, there were slight deviations from this correlation: beta-hydroxybutyric acid and the amino acids leucine, threonine, and L-hydroxyproline had higher normalized growth rates than expected based on their normalized carrying capacities, while N-cetylneuraminic acid, mannose, arbutin, salicin, arabitol, and sorbitol had lower normalized growth rates than expected.

## *C. difficile* lineages are distinguished by phylogenetically conserved metabolic capabilities

Because *C. difficile* is a genetically diverse species with many ribotypes, we wondered if ribotypes cluster based on their carbon substrate utilization profiles. We performed a principal component analysis of the normalized carrying capacities for the top 26 carbon substrates. Isolates strongly clustered by phylogenetic clades for the first two principal components, which collectively explained 56.9% of the variance (Fig. 2A and B), and showed clustering by ribotype for the first four principal components, which collectively explained 73.7% of the variance (Fig. 2C and D). A similar principal component analysis using the normalized growth rates for the top 26 carbon substrates detected similar but weaker clustering patterns (Fig. S2). Principal component loadings, which describe how substrates contribute to each principal component, showed that the clustering of both clades and ribotypes was mostly driven by nine substrates (Fig. 2E and F). These substrates were the sugars trehalose, mannose, ribose, and melezitose; the sugar derivatives arbutin and salicin; and the sugar alcohols sorbitol, mannitol, and arabitol. As expected, normalized carrying capacities on these substrates were highly variable between isolates (see coefficients of variation in Fig. 1). The direction of loadings also aligned with the growth of ribotypes on each substrate (Fig. S3). For example, trehalose supported the highest carrying capacities for isolates in Clades 2 and 5, and these are the isolates most positively correlated with the trehalose loading on principal component 2. Likewise, Clade 5 and ribotype 015 isolates are unable to grow on melezitose and occupy the principal component space opposite the direction of the melezitose loading on the first two principal components. In summary, genetically diverse isolates cluster by phylogenetic clade and ribotype based on their carbon substrate utilization.

### Emerging ribotypes have divergent metabolic phenotypes

To determine how substrate utilization differentiates ribotypes, we compared the growth of ribotypes on carbon substrates that supported *C. difficile* growth. First, we grouped isolates into sets based either on ribotype or habitat (e.g., lab-adapted). For each substrate, isolates were then ranked based on their normalized carrying capacity. Next, we used strain set enrichment analysis (SSEA) (an approach akin to gene set enrichment analysis [GSEA]; see Materials and Methods) to compute an enrichment score that reflects the degree to which a strain set (group of isolates) is overrepresented at the extremes (top or bottom) of the entire ranked list of isolates. Our analysis focused on groups that were profiled with a minimum of four isolates and included the following groups: RT027+, RT014+, RT106+, RT255, RT023, RT017, RT078, and Clade 5+ (i.e., non-RT078 isolates), and lab-adapted strains (CD630, R20291, VPI 10463, and M68). Here, a "+" sign indicates that a group included closely related ribotypes. Our analysis showed

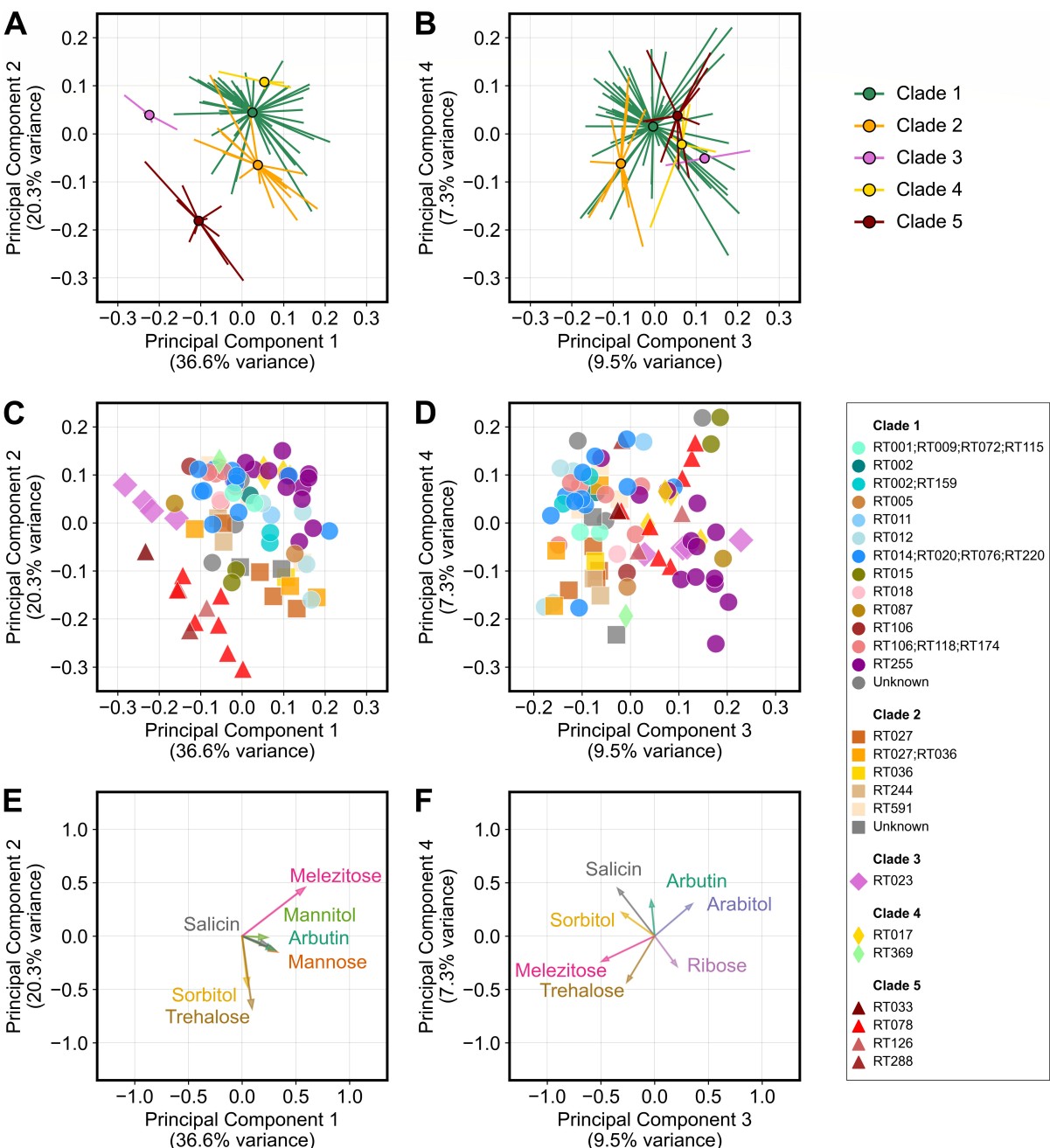

**FIG 2** Isolates cluster by phylogenetic clade and molecular ribotype based on their growth on carbon sources. (A and B) Top panels display the principal component analysis of the growth of isolates on the top 26 carbon substrates. Isolates are grouped by clades with circles indicating the centroid for each group and lines pointing to the location of each isolate in the ordination plots. (C and D) Middle panels display the same principal component analysis and further delineate the ribotype of each isolate with colors and clade with shapes, as indicated in the legend on the right. (E and F) Bottom panels visualize how certain substrates contribute to the position of isolates on the principal components (loading factors). Left column displays analysis for the first and second principal components, while the right column displays analysis for the third and fourth principal components.

that ribotypes 255, 027, and 017 are positively enriched (higher carrying capacities), while ribotypes 014–020, 106, and 023 are negatively enriched (lower carrying capacities) (Fig. 3). Clade 5 ribotypes, including ribotype 078, have a similar balance of positive and negative enrichments. By focusing only on significant enrichments, we surprisingly observed that ribotype 255 was the most positively enriched ribotype, while ribotype 023 was the most negatively enriched (Fig. 3; Fig. S4A), with a total of 14 substrates that

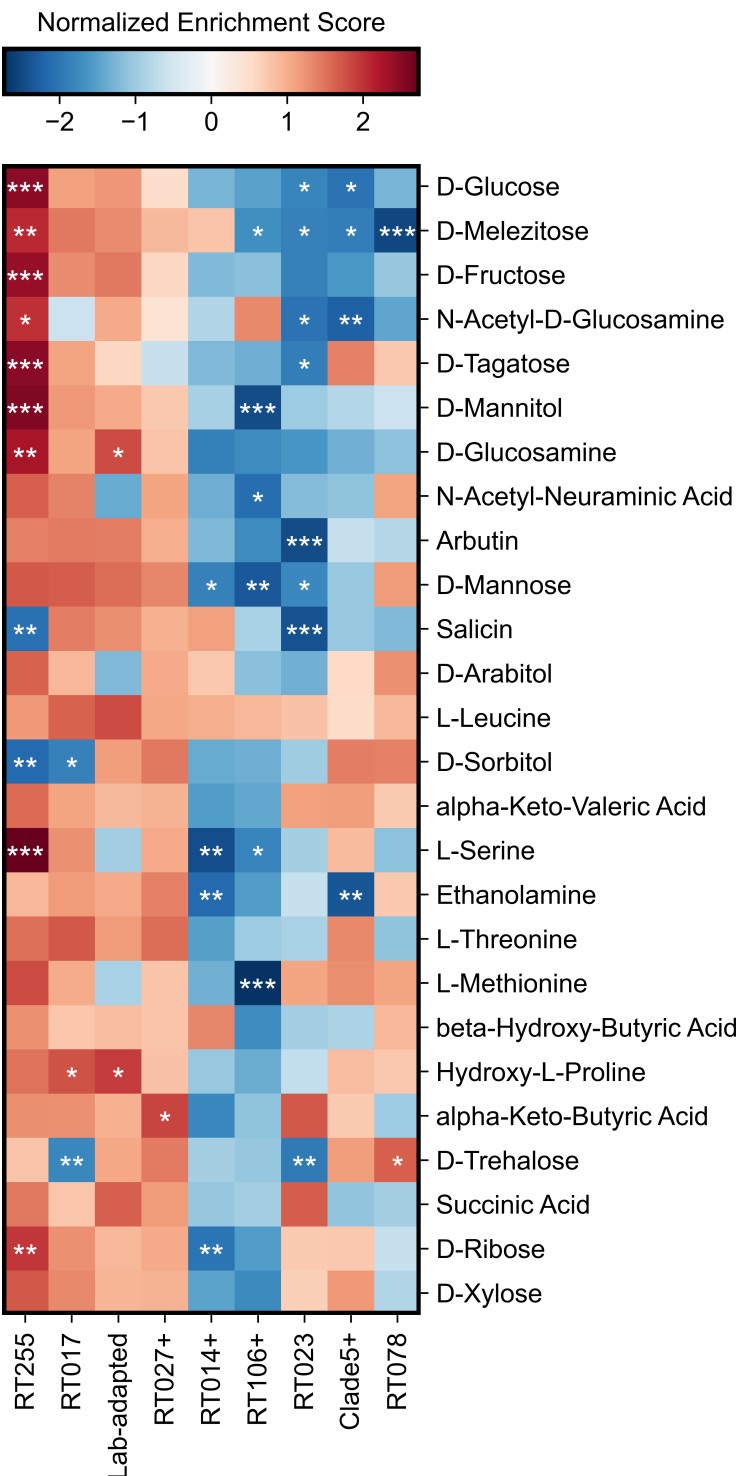

**FIG 3** Emerging and epidemic *C. difficile* ribotypes exhibit unique metabolic phenotypes. Heatmap displays which ribotypes are positively or negatively enriched for growth on each of the top carbon sources. Normalized enrichment scores were computed using strain set enrichment analysis based on normalized carrying capacities. Strain groups (columns) are hierarchically clustered based on similarity of their normalized enrichment scores, while substrates (rows) are ordered from top to bottom based on the median growth of all isolates, as shown in Fig. 1. For each substrate, statistical significance was estimated with a permutation-based test procedure and corrected with the Benjamini-Hochberg method. *P < 0.05, **P < 0.01, ***P < 0.001.

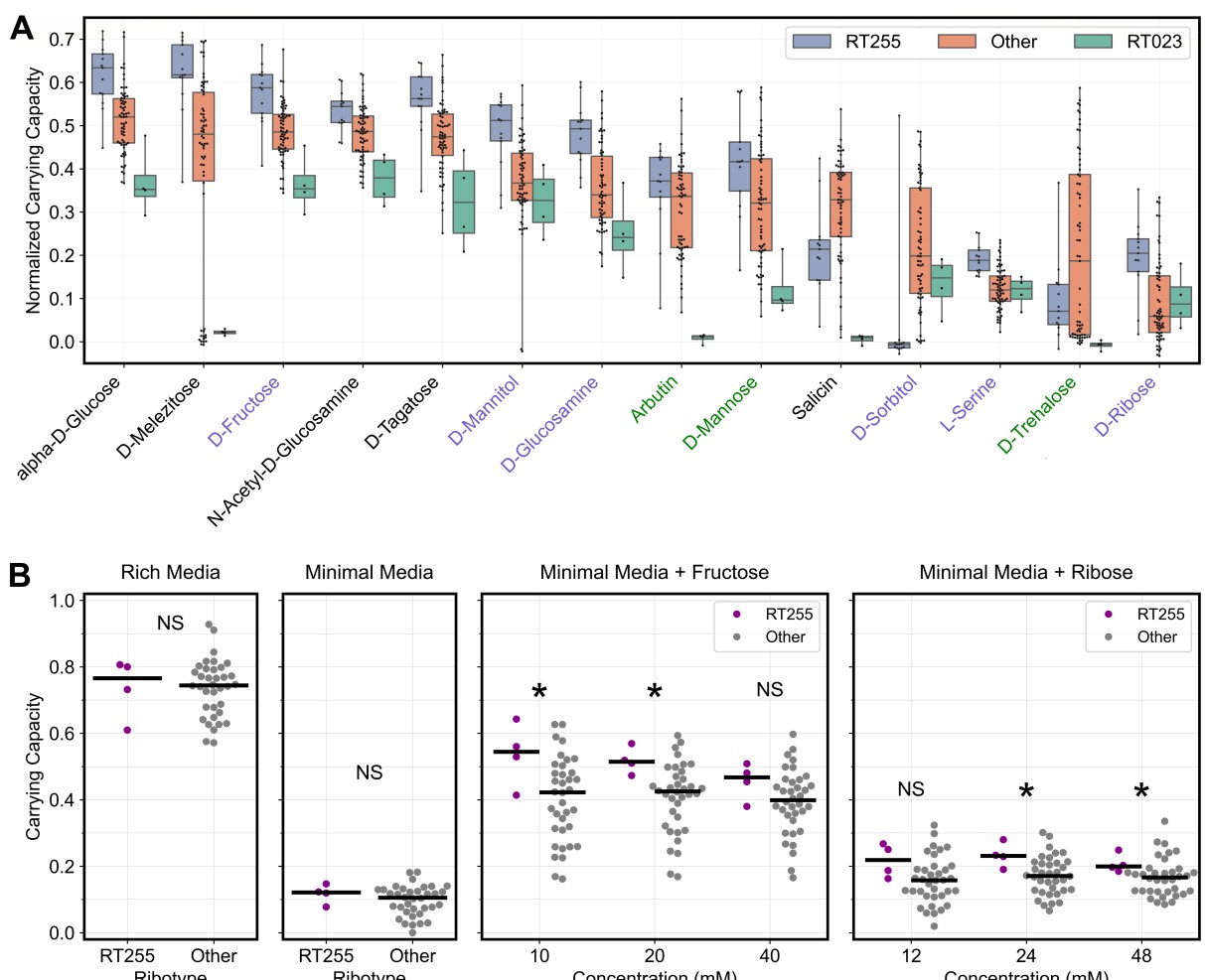

**FIG 4** Emerging *C. difficile* ribotypes have divergent metabolic phenotypes. (A) Plots compare the normalized carrying capacity for ribotype 255 and 023 isolates to other isolates on the 14 substrates that were significantly associated with these two emerging ribotypes using strain set enrichment analysis (see RT255 and RT023 columns in Fig. 3). Substrates that were significantly associated with both ribotypes were labeled with black text, whereas substrates that were significantly associated with only ribotype 255 or 023 were labeled with blue and green text, respectively. Whiskers span the range of each distribution from lowest to highest value. (B) In additional validation experiments, RT255 isolates reached significantly higher carrying capacity on fructose and ribose than isolates belonging to other ribotypes. *C. difficile* isolates harvested during exponential growth were inoculated into minimal media supplemented with either fructose or ribose at the indicated concentrations. As controls, we display the growth of isolates on minimal and rich media. Each data point represents the median of technical duplicates. *$P < 0.05$, **$P < 0.01$, ***$P < 0.001$, one-sided Wilcoxon rank-sum test with false discovery rate correction. NS indicates non significant differences.

were significantly associated with either or both of these emerging ribotypes (Fig. 4A). Comparison of the total growth of these ribotypes on these 14 substrates displayed a polarized enrichment pattern: Ribotypes 255 and 023 displayed very high or very low growth, respectively, in comparison to other isolates in the study (Fig. 4A). Furthermore, we performed a similar enrichment analysis but using normalized growth rates, and we detected similar patterns but fewer significant enrichments overall (Fig. S4B and S5). This divergence in enrichments suggests that a ribotype can emerge even if it grows poorly on many of the carbon sources commonly used by *C. difficile*.

## Emerging ribotype 255 grows robustly on a variety of substrates

Ribotype 255 is an emerging *C. difficile* lineage that comprised approximately 2–5% of cases in the United States between 2016 and 2018 (21–23). We observed that ribotype 255 was significantly enriched on various substrates (Fig. 3). These substrates included

fructose, which is a common sugar in the large intestine because of high dietary intake in the Western world (24, 25), and ribose, which had the second-highest coefficient of variation in the Biolog assays (Fig. 1). To verify these observations, we re-tested the growth of ribotype 255 and other isolates representing various common ribotypes on minimal media supplemented with fructose or ribose at multiple concentrations. Unlike Biolog assays, where cultures were started with inocula harvested from overnight cultures, we started cultures from inocula harvested during exponential growth and monitored growth for 24 h. As expected, we observed significantly higher growth for ribotype 255 on fructose and ribose ($P < 0.05$, one-sided Wilcoxon rank-sum tests with false discovery rate correction; Fig. 4B). We repeated this assay with fewer isolates but more substrates. The order of ribotypes on fructose, mannitol, salicin, and ribose recapitulated the patterns detected by our Biolog assays (Fig. S6). Ribotype 255 grew to a higher density than ribotypes 027, 106, and 014 on fructose, mannitol, and ribose, while ribotype 255 had the worst growth on salicin. Notably, although overnight cultures of ribotypes 017 and 078 were unable to grow on ribose in fresh media (Fig. S3), these ribotypes grew on ribose when it was introduced during exponential growth.

## Clades 1–4 and Clade 5 ribotypes display similar growth on simple sugars

Clade 5 is the most evolutionary distant *C. difficile* clade and includes ribotypes that are frequently detected in animals (26, 27). Comparative genomics suggested that the divergence of Clades 1–4 from Clade 5 is linked to the positive selection of genes involved in the metabolism of simple sugars (28). Therefore, we expected major metabolic differences between Clades 1–4 and Clade 5, but we only observed modest differences in growth between these clade groups. Animal-associated ribotypes (Clade 5+) were also distinguished only by increased growth on trehalose and sorbitol and decreased growth on glucose, N-acetyl-glucosamine, ethanolamine, and melezitose (Fig. 3). To further investigate these observations, we compared the growth of Clade 5 ribotypes to Clades 1–4 ribotypes on eight different substrates including the simple sugars glucose, fructose, tagatose, and ribose (Fig. 5). Cultures were started with mid-exponential growth inocula, and growth was monitored for 48 h. For Clade 5 isolates, we included both human- and animal-associated ribotype 078 strains and animal-associated ribotype 033, 126, and 288 strains. Based on the positive selection of genes involved in the metabolism of simple sugars (28), we hypothesized that Clades 1–4 would reach higher total growth on simple sugars. However, we did not detect any statistically significant differences between Clade 5 and other isolates on the eight tested substrates ($P > 0.05$, Student's $t$ test with false discovery rate correction). In summary, Clade 5 ribotypes grew robustly on simple sugars to a comparable level as Clades 1–4 ribotypes.

## Gene-environment interactions are necessary for growth on certain substrates

Thus far, we validated several aspects of our broad survey of carbon substrate utilization, including the heightened growth of Ribotype 255 and the comparable growth of Clades 1–4 and Clade 5 on various simple sugars. However, our survey also identified growth patterns that did not completely agree with prior work. In particular, we previously showed that ribotypes 027, 078, and 017 encode genetic variants or operons that facilitate competitive growth on trehalose (7, 29). Here, using our Biology assay, we confirmed that ribotypes 027 and 078 indeed grow robustly on trehalose but observed that ribotype 017 isolates do not. So, we re-tested M68, a ribotype 017 reference strain, on Biolog Phenotype MicroArray plates using different growth media for its overnight culture, before dilution and inoculation into Biolog plates. M68 was able to grow on trehalose when its inoculum was cultured overnight on brain heart infusion (BHI) media supplemented with high (5%) but not low (0.5%) yeast extract (Fig. 6). We hypothesized that high yeast extract in overnight culture provided a limiting growth factor for trehalose metabolism by ribotype 017. To test this, we reasoned that cell cultures that are

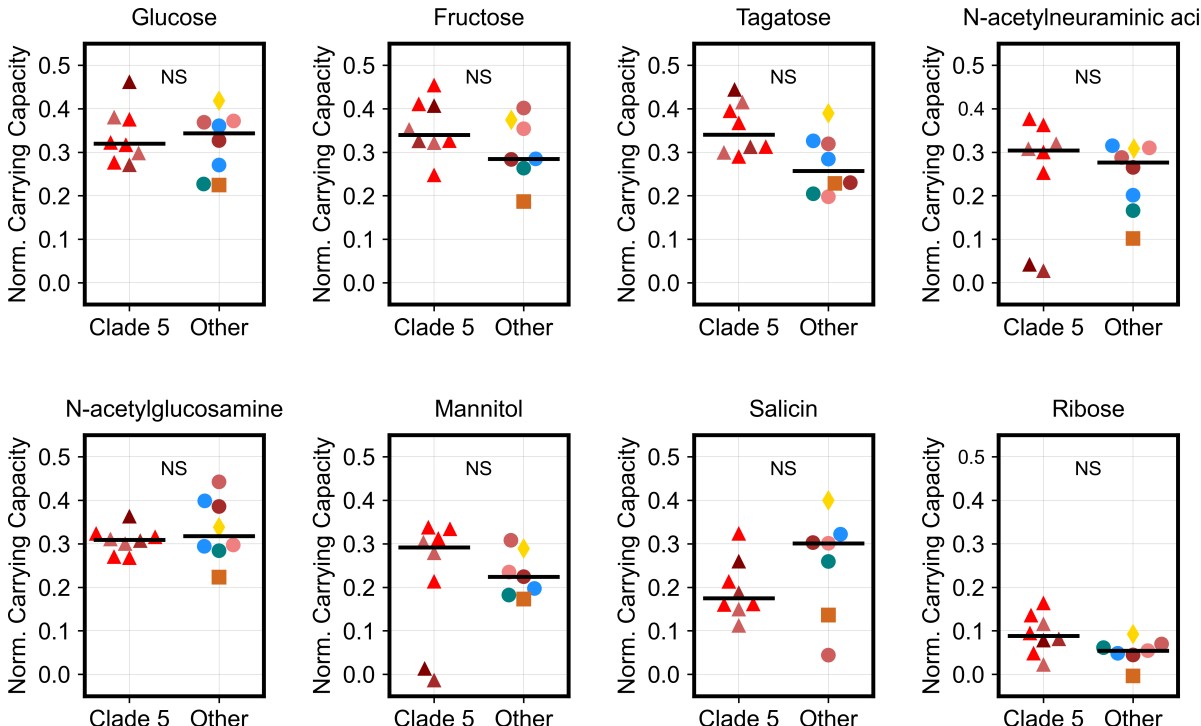

**FIG 5** Clade 5 ribotypes grow robustly on simple sugars similar to Clades 1–4 ribotypes. *C. difficile* isolates harvested during exponential growth were inoculated into minimal media supplemented with one of eight different carbon substrates, including the simple sugars glucose, fructose, tagatose, and ribose. Strip plots visualize the normalized carrying capacity or the additional growth for each isolate on each carbon source beyond baseline growth on minimal media. Colors and markers indicate the ribotype and clade of each isolate as shown in Fig. 2. Clade 5 isolates included ribotypes 078, 033, 126, and 288. Additional isolates tested belonged to ribotypes 001, 002, 014–020, 017, 027, and 053. Substrates were ordered from left to right by median growth of all isolates, as shown in Fig. 1. Each data point represents the median of technical duplicates. We did not detect any significant differences between Clade 5 and other clades on each of the tested substrates ($P > 0.05$, Student's $t$ test with false discovery rate correction). NS indicates non significant differences.

either washed or highly diluted before inoculating into minimal media would not retain the unknown growth factor and would not be able to grow on trehalose. Indeed, we observed that M68, which was pre-grown overnight on BHI with high yeast extract, was unable to grow on trehalose as the sole carbon source if it had been washed or highly diluted before inoculation (Fig. S7). In contrast, CD2015, a ribotype 027 isolate, was able to grow on trehalose even after both washing and dilution (Fig. S7). Altogether, this precarious growth of M68 suggests that an unknown limiting factor can modify the growth of some *C. difficile* strains on trehalose.

## DISCUSSION

By identifying the metabolic needs and preferences of *C. difficile*, we can inform the design of microbial and dietary interventions for the prevention and treatment of *C. difficile* infections. Indeed, microbial communities can be rationally designed to resist *C. difficile* by blocking access to key nutrients (30, 31), and dietary alterations can prevent enteric infections (32). Here, we expand our metabolic understanding of *C. difficile* by profiling carbon substrate utilization of clinical isolates comprising all five major clades. Our results corroborate that *C. difficile* is a bacterial generalist that can grow on a wide array of carbon sources, and that common *C. difficile* lineages exhibit unique metabolic capabilities. Surprisingly, we observed divergent metabolic profiles for emerging ribotypes 023 and 255, and robust growth of the evolutionarily distant Clade 5 on simple sugars. Altogether, our findings underscore the importance of considering the metabolic diversity of *C. difficile* in the design of microbial and dietary interventions.

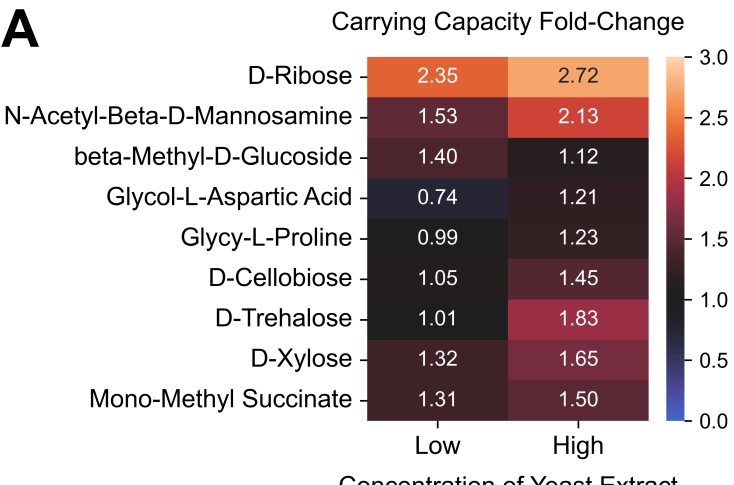

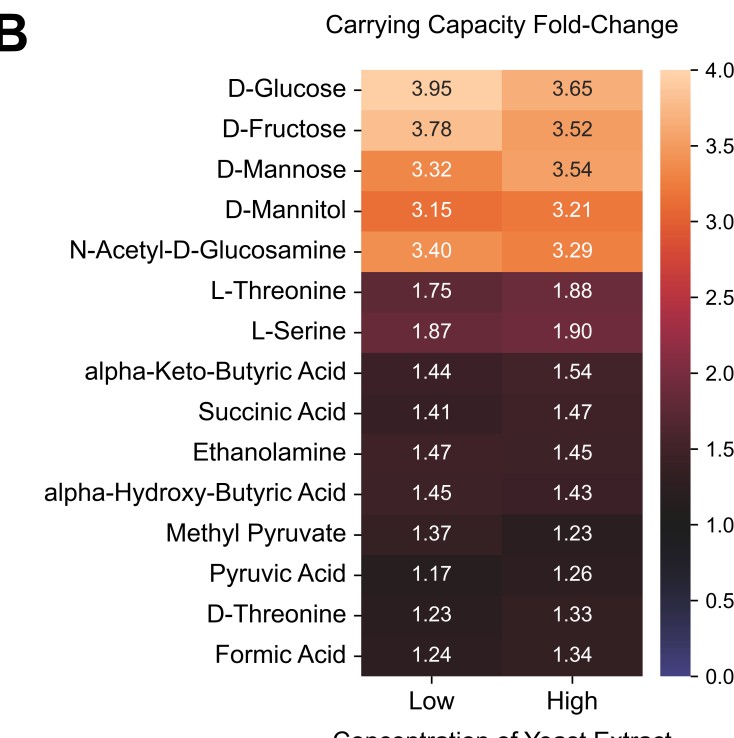

**FIG 6** Ribotype 017 isolate grows on trehalose and cellobiose only if pre-cultured on media with high yeast extract. *C. difficile* strain M68 was profiled for growth in a Biolog PM1 plate after pre-culturing overnight in brain heart infusion with either low (0.5%) or high (5%) yeast extract. Heatmaps display the fold-change for carrying capacity on minimal media supplemented with each substrate relative to carrying capacity on minimal media only; a fold-change of one therefore indicates no growth. (A and B) All displayed substrates supported the growth of M68 with fold-change values of at least 1.2 on either or both of the yeast extract concentrations. (A) Heatmap displays only the substrates with fold-change values for growth on high yeast extract that were at least 10% higher or lower than fold-change values for growth on low yeast extract, while (B) heatmap displays the remaining substrates. For each heatmap, substrates were ordered by agglomerative hierarchical clustering of fold-change values using UPGMA.

Our results also highlight several carbohydrates that are variably used by *C. difficile* lineages and may be crucial in the design of dietary interventions. These substrates

include trehalose, sorbitol, and ribose, which were associated with high coefficients of variation in our Biolog assay and explained significant variation in the clustering of isolates by their carbon substrate utilization. We and others have previously shown that trehalose metabolism is variable between ribotypes and that increased use of trehalose in food manufacturing correlated with the emergence of ribotypes that are highly adept at consuming trehalose (7, 29, 33, 34). In contrast, sugar alcohols are derived in the intestines from both host metabolism and diet. In mouse models of *C. difficile* infections, sugar alcohols were highly abundant in the guts of susceptible animals (35), and sorbitol utilization genes were highly upregulated during *C. difficile* toxin-induced inflammation (12). Because sorbitol comprises the most used non-nutritive sweetener globally (25, 36), it is likely more common in human diet than trehalose. Ribose is also highly abundant in the gut because it comprises the backbone of nucleobases. Importantly, gut commensals have adapted to sense and scavenge ribose from dietary nucleosides (37) and ribose transport and metabolism are highly expressed in mice mono-colonized with *C. difficile* (38). Nonetheless, additional studies are needed to verify the role of these nutrients in *C. difficile* colonization and pathogenesis.

By profiling the growth of all five major clades, our study also tested prevailing hypotheses about the evolutionary adaptation of the *C. difficile* species. A large-scale genomic analysis suggested that the divergence of Clades 1–4 from Clade 5 is linked to the metabolism of dietary simple sugars, for example, by identifying positive selection on genes involved in fructose metabolism, glycolysis, sorbitol, and ribulose metabolism (28). A related mouse experiment also showed that fructose and glucose could differentially impact the colonization of *C. difficile* Clades 1–4 (28). However, this mouse experiment was biased by testing only two strains, Clade 2 R20291 and Clade 5 M120, which are both lab-adapted. In contrast, we showed that ancestral Clade 5 isolates are just as capable as newer Clades 1–4 isolates in terms of growth on simple sugars. Further, the two emerging lineages, ribotypes 255 and 023, which are in Clades 1 and 3, respectively (22, 39), had divergent metabolic phenotypes, especially in terms of growth on simple sugars. Yet, there are several potential biological explanations for higher colonization of R20291, in contrast to M120, in mice fed simple sugars, including non-regulatory factors, sporulation, toxin production, or sugar-microbiome interactions. So, while our findings highlight the need for more representation of a pathogen's diversity in animal studies, it remains possible that simple sugars differentially impact the growth of newer *C. difficile* lineages in the gut.

Our results also emphasize that genomic potential does not always translate to metabolic activity. We previously showed that certain ribotypes can utilize trehalose at ultra-low concentrations due to unique gene variants or operons (7, 29). These genetic signatures were also shown to be encoded in additional ribotypes (40), but it is unclear if the presence of these gene variants is sufficient for growth on trehalose. For example, while ribotype 023 lacks the canonical *treRA* operon, it encodes an alternative four-gene operon (40, 41) that grants ribotype 078 (and other Clade 5 isolates) the ability to grow on 10 mM trehalose (7). Yet, we were unable to grow ribotype 023 isolates even at higher concentrations of trehalose, possibly because of a truncation in *treX* (41). Similarly, we have previously shown that ribotype 017 isolates can grow on trehalose (29), but replication of this growth, in this study, required an unknown limiting factor available in yeast extract. Because we only validated that this limiting factor supports the growth of M68 on trehalose, additional studies are needed to verify if growth factors in yeast extract also facilitate higher growth of other strains on trehalose and higher growth of M68 on other substrates, including cellobiose, ribose, and xylose.

Despite the evidence presented here, the metabolic capabilities of clinical *C. difficile* isolates still need to be further studied. Our analysis was limited to the 190 carbon substrates included in the Biolog Phenotype MicroArray plates. Also, these substrates are seeded in these plates at unknown proprietary concentrations which may result in missing interesting patterns, including the ability of certain ribotypes to grow on low levels of substrates such as trehalose (7). Moreover, we profiled growth using a specific

growth protocol, and results may not generalize to different experimental conditions. For example, Clade 5 isolates were able to robustly grow on ribose if exposed to it during mid-exponential growth but not after overnight growth. In addition, Stickland fermentation of amino acids is crucial for *C. difficile* pathogenesis (8, 42–44), but our assay does not adequately discern the growth contribution of amino acids in the Biolog plates from amino acids in the minimal media. Although amino acids were the sole nitrogen source in our minimal media, they were also fermented as carbon sources by *C. difficile*. A defined growth medium that minimizes amino acids is necessary to properly investigate Stickland fermentation, as has been developed for the study of the Wood-Ljungdahl pathway in *C. difficile* (18). Finally, *in vitro* growth of *C. difficile* on individual carbon sources does not reflect how *C. difficile* interacts with the complex metabolic diversity available in the human gut, nor does it address other factors that impact *C. difficile* colonization in the human gut, including sporulation, toxin production, and interaction with other gut microbes.

Ultimately, our comprehensive study of carbon source utilization shows that clinical *C. difficile* isolates are bacterial generalists with lineage-specific metabolic adaptations. It remains unclear if these adaptations have contributed to the emergence of certain lineages or if they provide a competitive advantage in the human gut. Further, our results suggest the existence of unknown factors that are necessary for the growth of specific *C. difficile* lineages in certain nutritional environments. The discovery of these growth factors presents new opportunities to understand and control the engraftment of *C. difficile.* Finally, our analytical approach for comprehensively profiling the metabolic capacities of *C. difficile* can be adapted for the study of additional pathogens to discover insights about their metabolic needs in the gut.

## MATERIALS AND METHODS

### Bacterial strains

Clinical *C. difficile* strains were isolated from a variety of sources, including patients in the United States ($n = 62$), the Netherlands ($n = 6$), Colombia ($n = 5$), Japan ($n = 4$), Australia ($n = 2$), the Czech Republic ($n = 1$), Spain ($n = 1$), and the United Kingdom ($n = 1$). Lab-adapted *C. difficile* strains included R20291, CD630, VPI 10463, and M68. Table S1 and S2 describe the source and molecular typing of these isolates, respectively. Notably, ribotype 255 strains were isolated from patients in the state of Texas by Kevin Garey's lab, while ribotype 023 strains were isolated from patients in various European sites by the labs of Ed Kuijper and Brendan Wren.

### Bacterial growth assays

All preparation of bacterial strains and growth of bacterial cultures were performed under an anaerobic atmosphere (5% $CO_2$ , 5% $H_2$, and 90% $N_2$) inside a vinyl anaerobic chamber (Coy Laboratory Products). Bacterial strains were revived from frozen stocks by streaking on brain heart infusion agar supplemented with 0.5% yeast extract, and then growing at 37°C.

### Biolog phenotype microarray assays

Colonies on agar plates were cultured overnight for approximately 15–18 h in BHIS broth, which is brain heart infusion media supplemented with 5% yeast extract. Cell cultures were then diluted 1:10 in a defined minimal media, which is a basal defined medium adapted from table 1 of Karasawa et al. (45) with several adjustments (Table S3). Each well in the pre-reduced Biolog Phenotype MicroArray plates was then filled with 100 µL of the diluted cell cultures. Plates were sealed with gas-permeable film and then incubated in a microplate reader (Sunrise by Tecan Life Sciences or AccuSkan by Thermo Scientific) for 17 h. Optical density (620 nm) was measured every 10 min immediately after 5 s of orbital shaking. The growth profiles for isolates were repeated at least two

times except for two isolates, a Ribotype 106 isolate and another of unknown ribotype, which were measured only once.

## Bacterial growth validation assays

Overnight cultures in BHIS broth were subcultured 1:5 in fresh media and grown to mid-exponential phase (OD ~0.6), then diluted 1:50 in defined minimal media. Cell suspensions were mixed 1:1 with twice-concentrated substrate solutions, which were prepared in double-deionized water and then filter-sterilized (0.22 µm pore size). Each well had a final volume of 200 µL of cell cultures, which were grown statically for 24 hs in a microplate reader (Sunrise by Tecan Life Sciences). Optical density (620 nm) was measured every 10 min immediately after 5 s of orbital shaking. For validation of M68 and CD2015 growth on trehalose, overnight cultures were washed inside the anaerobic chamber by spinning at 2,000 g for 2 min with a pulse centrifuge, decanting the supernatant, and then resuspending pellets in defined minimal media.

## Microbial growth curve analysis

The "AMiGA" software (https://github.com/firasmidani/amiga) was used for the analysis of growth curves collected in this study (17). To flag wells with growth issues, we used the "summarize" command by AMiGA to generate 96-well growth curve plots. We manually flagged wells where growth curves displayed either rapid spikes or dips in optical density (often caused by gas bubbles), high background noise (as determined by starting OD), or unusual noisy fluctuations in optical density. Plates that included a large number of flagged wells were excluded from analysis. Because growth curves of *C. difficile* do not always follow the classical logistic growth dynamics, we modeled growth curves using Gaussian Process regression with the "fit" command by AMiGA, which estimated growth metrics including area under the curve, carrying capacity, and growth rates. Because *C. difficile* can grow in defined minimal media using Stickland fermentation of amino acids, we wanted to compare the growth dynamics of each *C. difficile* isolate on different substrates relative to its own growth on minimal media. Therefore, we computed the difference in each growth metric relative to its median value during growth on minimal media, either using the "normalize" command by AMiGA or custom-written Python script.

## Principal component analysis

To see if isolates cluster by molecular ribotype or phylogenetic identity, we performed principal component analysis on the normalized carrying capacity for the top 26 substrates used by *C. difficile*. These top substrates increased the carrying capacity by an OD of 0.1 (relative to carrying capacity on minimal media) for at least 10% (i.e., 8) of the tested *C. difficile* isolates. Principal component analysis was performed using the "statsmodels" Python package on zero-centered normalized carrying capacities. Using principal component loadings, we compared the contribution of each substrate to the position of isolates on each principal component. We visualized the loadings only for substrates that were ranked in the top five in terms of loadings for each of the first four principal components.

## Strain set enrichment analysis

We devised a statistical approach that we termed SSEA in order to identify whether a *C. difficile* lineage can use certain substrates better or worse on average than other lineages. SSEA is methodologically similar to GSEA. Whereas GSEA tests how sets of genes are enriched or depleted in terms of expression on a specific experimental condition (46), SSEA tests how sets of isolates (e.g. clades, ribotypes, or other arbitrary grouping) are enriched or depleted in terms of normalized carrying capacity or normalized growth rate on a specific substrate. SSEA was implemented using the "prerank" function by the "gseapy" Python package (47). For this analysis, we only included sets with a minimum

size of 4, weighted enrichment scores by a value of 0.5, and estimated *P* values with 10,000 permutations. *P* values were adjusted with Benjamini-Hochberg false discovery rate correction.

## Statistical tests

To statistically test the effect of different substrates and molecular identities on the growth of *C. difficile* isolates, we used Wilcoxon rank-sum tests and Student's *t* tests using the "stats" R package. Prior to any of these tests, we calculated the median value for each group of technical replicates in the data set. False discovery rate correction was performed using the Benjamini-Hochberg method.

## ACKNOWLEDGMENTS

The authors would like to thank the following for collecting or providing *C. difficile* isolates that were profiled in this study: Ed Kuijper (Leiden University), Brendan Wren (London School of Hygiene and Tropical Medicine), Angel Augusto Gonzalez Marin (Universidad de Antioquia), Haru Kato and Mitsutoshi Senoh (National Institute of Infectious Diseases, Tokyo, Japan), Sara McNamara (Michigan Department of Community Health), Joseph Sorg (Texas A&M University), and Dena Lyras (Monash University). The authors also thank Lei Pan and James Collins for processing six strains with Biolog phenotype microarray plates and Eva Preisner for sharing related data.

This research was supported by several NIH grants: F.S.M acknowledges support from T32DK007664, H.A.D. acknowledges support from F32AI136404, and R.A.B. acknowledges support from U19AI157981, R01AI123278, and U01AI124290. Data analysis was performed on the HPC cluster that is managed by the Biostatistics and Informatics Shared Resource (BISR) and supported by an NCI P30-CA125123 and Institutional funds from the Dan L Duncan Comprehensive Cancer Center and Baylor College of Medicine.

F.S.M. and R.A.B. conceptualized the study. F.S.M. and H.A.D. administered the study. F.S.M., H.A.D., and R.A.B. developed methods. K.W.G. and J.K.S. provided bacterial strains. J.K.S. sequenced isolates. F.S.M., H.A.D., N.M., and C.K.A. performed growth experiments. F.S.M. and N.M. performed validation experiments. F.S.M. developed software, analyzed data, curated data, and visualized results. R.A.B. and K.W.G. acquired funding. R.A.B. supervised the project. F.S.M. wrote the manuscript. All authors reviewed and edited the manuscript.

## AUTHOR AFFILIATIONS

[1]Alkek Center for Metagenomics and Microbiome Research, Baylor College of Medicine, Houston, Texas, USA
[2]Department of Molecular Virology and Microbiology, Baylor College of Medicine, Houston, Texas, USA
[3]Department of Pharmacy Practice and Translational Research, University of Houston, Houston, Texas, USA
[4]Department of Pathology and Immunology, Baylor College of Medicine, Houston, Texas, USA

## AUTHOR ORCIDs

Firas S. Midani  http://orcid.org/0000-0002-2473-7758
Heather A. Danhof  http://orcid.org/0000-0002-6347-0021
Nathanael Mathew  http://orcid.org/0000-0003-4695-4005
Colleen K. Ardis  http://orcid.org/0000-0003-3866-2897
Kevin W. Garey  http://orcid.org/0000-0003-2063-7503
Jennifer K. Spinler  http://orcid.org/0000-0002-7830-7665
Robert A. Britton  http://orcid.org/0000-0001-8983-9539

## FUNDING

| Funder | Grant(s) | Author(s) |
|---|---|---|
| HHS | National Institutes of Health (NIH) | T32DK007664 | Firas S. Midani |
| HHS | National Institutes of Health (NIH) | F32AI136404 | Heather A. Danhof |
| HHS | National Institutes of Health (NIH) | U19AI157981 | Robert A. Britton |
| HHS | National Institutes of Health (NIH) | R01AI123278 | Robert A. Britton |
| HHS | National Institutes of Health (NIH) | U01AI124290 | Robert A. Britton |

## AUTHOR CONTRIBUTIONS

Firas S. Midani, Conceptualization, Data curation, Formal analysis, Investigation, Methodology, Project administration, Software, Validation, Visualization, Writing – original draft, Writing – review and editing | Heather A. Danhof, Investigation, Methodology, Project administration, Writing – review and editing | Nathanael Mathew, Investigation, Validation, Writing – review and editing | Colleen K. Ardis, Investigation, Writing – review and editing | Kevin W. Garey, Funding acquisition, Resources, Writing – review and editing | Jennifer K. Spinler, Investigation, Resources, Writing – review and editing | Robert A. Britton, Conceptualization, Funding acquisition, Methodology, Resources, Supervision, Writing – review and editing

## DATA AVAILABILITY

Microplate reader data are available at https://doi.org/10.5281/zenodo.12626877. Code for analyzing data and creating figures is available at https://github.com/firasmidani/cdiff-biolog-growth. The full list and versions of computational tool used for data analysis are included in the GitHub repository.

## ADDITIONAL FILES

The following material is available online.

### Supplemental Material

**Figure S1 (mSystems01075-24-s0001.tiff).** Distribution of normalized growth rates by substrate.
**Figure S2 (mSystems01075-24-s0002.tiff).** Principal component analysis of normalized growth rates.
**Figure S3 (mSystems01075-24-s0003.tiff).** Normalized carrying capacities by substrate and clade.
**Figure S4 (mSystems01075-24-s0004.tiff).** Summary of strain set enrichment analysis.
**Figure S5 (mSystems01075-24-s0005.tiff).** Strain set enrichment analysis of normalized growth rates.
**Figure S6 (mSystems01075-24-s0006.tiff).** Growth validation experiment on four substrates.
**Figure S7 (mSystems01075-24-s0007.tiff).** Experiment for limiting factor in yeast extract.
**Supplemental legends (mSystems01075-24-s0008.docx).** Legends for supplemental figures and tables.
**Supplemental tables (mSystems01075-24-s0009.xlsx).** Tables S1 to S3.

### Open Peer Review

**PEER REVIEW HISTORY (review-history.pdf).** An accounting of the reviewer comments and feedback.

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
