## [Reviewer comments · mSystems]

Emerging *Clostridioides difficile* ribotypes have divergent metabolic phenotypes

Firas Midani, Heather Danhof, Nathanael Mathew, Colleen Ardis, Kevin Garey, Jennifer Spinler, and Robert Britton

Corresponding Author(s): Firas Midani, Baylor College of Medicine

Review Timeline:

Submission Date:	August 14, 2024
Editorial Decision:	September 17, 2024
Revision Received:	December 26, 2024
Accepted:	February 6, 2025

Editor: Christian Diener

Reviewer(s): Disclosure of reviewer identity is with reference to reviewer comments included in decision letter(s). The following individuals involved in review of your submission have agreed to reveal their identity: Jordy Evan Sulaiman (Reviewer #1); Joseph P Zackular (Reviewer #2)

Transaction Report:

DOI: <https://doi.org/10.1128/msystems.01075-24>

Re: mSystems01075-24 (Emerging *Clostridioides difficile* ribotypes have divergent metabolic phenotypes)

Dear Dr. Robert A Britton:

Both reviewers, and me as well, were enthusiastic about the manuscript and valued the large number of strains tested here. They did feel that some of the claims made in the manuscript would require some additional evidence to stand in its current form. Additionally, they also asked you to make some changes to the manuscript text in order to improve clarity/flow and discuss how the observed differences in carrying capacity fit in with the complex mechanisms of *C. difficile* infections in the human gut (such as interactions with the resident microbiome, toxin production, and sporulation). Please also see the detailed reviewer comments below. Thus, I would ask you to revise the manuscript accordingly.

Revision Guidelines

Sincerely,
Christian Diener
Editor
mSystems

Reviewer #1 (Comments for the Author):

In this paper, Midani et al. characterized the metabolic diversity of 88 *C. difficile* isolates (which are classified into 22 ribotypes). The authors found that *C. diff* is a generalist, capable of growing on a variety of carbon substrates. They showed that *C. diff* displays ribotype & clade-specific metabolic capabilities, and two emerging Ribotypes 023 and 255 have different metabolic phenotypes. The authors claimed that *C. diff* clade 5, the most evolutionary distant clade, displayed the most robust growth on simple sugars, which poses questions about the adaptation of *C. diff* to the human gut.

Overall, the authors did a good job in the Results section in explaining their analysis in detail. With the comprehensive metabolic characterization of a large number of *C. diff* isolates, this paper is a good contribution to the field. However, the authors need to tone down some of the claims made in the paper, especially on the implications of the study. Although the paper showed that *C. diff* ribotypes have distinct metabolic capabilities that could help them survive in diverse environmental conditions, the phenotype assessed in this study is simply monoculture growth on different single carbon sources, which is not physiologically relevant. It is difficult to draw any correlations on the adaptation of *C. diff* in the human gut (as claimed in the importance section). The authors did not analyze differences in toxin production which is a major determinant of *C. diff* infection, interactions with human gut microbiota, nor consider other *C. diff* life cycles (sporulation & germination). Discussion about this would be good to acknowledge the limitations and scope of the study. Finally, the author's claim regarding *C. diff* clade 5 exhibiting the most robust growth on simple sugars is not supported by the current data. Specific comments are listed below:

Introduction

1. In the introduction, the authors motivate the study by saying that diet could impact gut microbiota composition, modulate *C. diff* disease severity, and contribute to the emergence of epidemic lineages, but how human diet impacts *C. diff* epidemiology and pathogenesis is unclear. However, the paper did not use human diets, but rather simple sugars. Human diets consist of polysaccharides, which cannot be consumed readily by *C. diff*. There are inter-species interactions with the gut microbiota (e.g. fiber degraders with PULs) that mediate *C. diff* growth & colonization.
2. As written in the Introduction section, the fact that *C. diff* is a generalist species capable of growing in multiple substrates is not new. There are a lot of other previous studies that have shown this through growth assays on different substrates and also metabolomics experiments. This is perhaps not a suitable claim to be made in the abstract, or at least the authors should revise the claim that their study "supports the idea that *C. diff* is a generalist".
3. The author's claim in the introduction "it is unknown if clinical *C. diff* isolates have similar metabolic capabilities as lab-adapted strains." is not true. A recent study (<https://doi.org/10.1038/s41467-024-51062-w>) investigated the metabolic capabilities of 19 *C. diff* isolates and also a lab strain *C. diff* DSM 27147 (R20291) to study differences in inter-species interactions, which should be acknowledged. In this paper, the authors performed a more comprehensive metabolic assessment with a larger sample size (88 isolates) and focused on whether certain ribotypes have metabolic advantages over others, which should be the point they emphasize.
4. Lines 61: Referencing ref. 8 two times in the same sentence: "Because it is adept at fermenting both amino acids and carbohydrates (8, 8)".

Results

1. The authors should justify the use of Carrying capacity as a metric to measure *C. diff* growth and mention it in the text. In many laboratory media conditions, *C. diff* growth comprised of logistic growth followed by a highly variable stationary phase with a steep reduction in OD600.
2. Since the minimal media contain amino acids and some of them are major resources for *C. diff* growth, e.g. proline, the plots in Figure 1 are therefore the net change in *C. diff* growth from baseline growth under amino acids, which the Authors acknowledged. This sentence might not be true "*C. difficile* reached the highest biomass on simple sugars, followed by sugar derivatives, sugar alcohols, amino acids, then carboxylic acids" since the authors are comparing growth from a baseline growth under amino acids.
3. The fact that some resources (e.g. trehalose) show higher COV in growth between isolates is interesting and should be mentioned in the results. This could be because some ribotypes are better at utilizing trehalose than others (<https://doi.org/10.1073/pnas.2119396119>). This was observed in a previous paper from the authors (<https://doi.org/10.1073/pnas.2119396119>) and another study (<https://doi.org/10.1038/s41467-024-51062-w>), which should be cited. I think this is mentioned in the Discussion a bit in Paragraph 2 but also needs to be mentioned in the Results section. The resources with higher COV are likely the ones that appear in the principal component loadings in Fig. 2e-f that contribute to the different clusters of clades/ribotypes based on growth phenotypes.
4. Provide exact p values in Fig. 4b. Although the trend is there, it is unclear if the difference between RT255 & others is statistically significant from the plots.
5. Lines 147-148 & Figure 4b: Why fructose and ribose specifically? The authors should explain the rationale why they pick these two, since there are many other substrates where RT255 has higher growth (Fig. 3).

6. I am not convinced by the claim that "Clade 5 ribotypes grow more robustly on simple sugars than Clade 1-4 ribotypes". From Fig. 3, we see that Clade 5 has a negative enrichment score in many substrates. The experiments in Fig.5 were merely based on 8 substrates (with 4 simple sugars: glucose, fructose, tagatose, and ribose). The results show that Clade 1-4 and Clade 5 have similar growth overall ($P > 0.05$) and Clade 5 grew to a higher density only on two sugars (ribose and tagatose). This does not support the author's claim "Clade 5 ribotypes grew more robustly on simple sugars than common ribotypes in Clades 1-4". The authors may need to do more validation experiments with more simple sugars to be able to draw a clear conclusion.

7. "Altogether, this precarious growth of M68 suggests that an unknown limiting factor can modify the growth of some *C. difficile* strains on certain substrates, including trehalose, ribose, and cellobiose." Where is the validation data for ribose and cellobiose? The authors only provide validation experiments for trehalose in Fig. S5

Discussion

1. Paragraph 2: It is good that the authors discuss how the sugars they used in this study could be relevant in human diets, but the authors should also acknowledge/discuss that these simple sugars may not be physiologically relevant, as human diets consist of polysaccharides (e.g. dietary fibers) which are not explored in this study.

2. The authors already discussed some of the limitations and the scope of the study in the second last paragraph. It will be good to incorporate other limitations mentioned above in this paragraph.

Reviewer #2 (Comments for the Author):

In this manuscript, Midani and colleagues profile carbon substrate utilization of a collection of *C. difficile* clinical isolates to establish links between carbon metabolism and evolution or emergence of *C. difficile* ribotypes. The authors report that *C. difficile* clinical isolates are generally capable of growth on many carbon sources. Carbon source preference clustered by both clade and ribotype, indicating phylogenetic conservation of metabolic capabilities. Importantly, they found that two emerging *C. difficile* ribotypes displayed differences in preferred carbon sources, suggesting that carbon metabolism is not a main driver of ribotype emergence. Overall, there is high enthusiasm for this work. These data provide clinical relevance to metabolic studies performed in laboratory strains of *C. difficile* and provide new insights into important clinically relevant strains of this important pathogen. Some minor comments below:

- The description of Figure 4 in the text is a bit confusing. Figure 4A is referenced, but not described in detail. In addition, the text does not reflect the fact that the metabolic phenotypes of RT023 were also experimentally confirmed, and instead the text focuses entirely on RT255. Also added context for importance of utilization of fructose and ribose by RT255 is needed.
- The transition to Figure 6 seemed abrupt and more rationale is needed. Namely, the introduction to trehalose utilization would be useful. For instance, what is the effect of the mentioned C171S amino acid substitution? Not clear without exploring previous literature that this mutation enhances utilization, not decreases.
- Experiments highlighted in Figure 6 and accompanying conclusions need added experimentation or description to provide insights into these observations. What factors in yeast extract are driving this phenomenon? Is there alteration in activity or expression of the trehalose utilization operon in high, low, and no yeast extract cultures? Is the trehalose utilization operon required for this observation? Do trehalose adaptation experiments (low level trehalose in overnights followed by growth in high levels of trehalose) provide similar results? It is noted that RT027 does not respond to washing, but there is a seemingly significant decrease in growth following washing.
- The authors note in their methods that they calculate area under the curve and growth rates, but these data are not included. In addition to carrying capacity, understanding how different strains growth dynamics differ under different conditions would be quite informative. Presumably ranking of substrates could be done with growth rate, as well? Was this analysis performed? If not, this should be noted as a limitation in the discussion, as carrying capacity isn't necessarily the most important factor in fitness/virulence.

Response to Reviewers

Both reviewers, and me as well, were enthusiastic about the manuscript and valued the large number of strains tested here. They did feel that some of the claims made in the manuscript would require some additional evidence to stand in its current form. Additionally, they also asked you to make some changes to the manuscript text in order to improve clarity/flow and discuss how the observed differences in carrying capacity fit in with the complex mechanisms of *C. difficile* infections in the human gut (such as interactions with the resident microbiome, toxin production, and sporulation). Please also see the detailed reviewer comments below. Thus, I would ask you to revise the manuscript accordingly.

We thank the reviewers for their reading and enthusiastic feedback on our manuscript #mSystems01075-24: Emerging *Clostridioides difficile* ribotypes have divergent metabolic phenotypes. We have carefully reviewed and addressed the comments by the reviewers. First, we would like to highlight some of the main changes to the manuscript.

Revised text to improve clarity and flow of the manuscript. We revised our introduction and discussion sections to clarify the motivation and focus of our study. Our revised manuscript is focused on identifying ribotype-specific metabolic differences, instead of the more complex diet-microbe interactions or differences between lab-adapted and clinical *C. difficile* isolates. Accordingly, our discussion section primarily focuses on the nutritional preferences or metabolic phenotypes of *C. difficile* but discusses the limitations of our work in terms of the more complex dietary and microbial factors that impact *C. difficile* colonization in the gut. In addition, we included additional rationale for why we selected fructose and ribose for validation in the experiment in Figure 4, and why we performed experiments in Figures 5 and 6.

Re-analysis to improve interpretation of the data and properly support our claims. Reviewers were concerned or unsure about the interpretation of some of our validation experiments (Figures 4 and 5). Therefore, we simplified our analysis by using univariate two-sample tests instead of multivariate linear models. These univariates test are more conservative and easier to interpret by the reader. For Figure 4, the re-analysis did not affect our claim or the interpretation of the results. For Figure 5, the re-analysis slightly affected the interpretation of the results. Previously, we suggested that Clades 5 ribotypes grow better on simple sugars than Clade 1-4 ribotypes. Our reanalyzed data suggest that these clades have similar growth on simple sugars. This is still a surprising finding because a recent large-scale genomics analysis (Reference #28) identified positive selection on genes involved in simple sugar metabolism for newer Clades 1-4. Yet, our results show that ancient Clade 5 and newer Clade 1-4 ribotypes have similar growth on simple sugars.

Additional figures on growth rates. In the initial submission, we focused on the total growth (or carrying capacity) of *C. difficile* isolates on the various substrates tested. In response to a reviewer's comments, we now also include additional analysis using growth rates for the top substrates consumed by *C. difficile* in our study (see Figures S1, S2, S4B and S5 which complement Figures 1, 2, S4A, and 3, respectively). Our additional analysis did not reveal markedly new insight about the growth dynamics of *C. difficile* ribotypes, which may suggest that metabolic adaptation within the *C. difficile* species mostly affects total growth and growth rate in a similar fashion.

Below, we have structured our point-by-point response by first including the reviewer's comment (in gray) and then our response (in black). In each response, line numbers indicate the location of these changes in the marked-up manuscript. For your convenience, we included any revised relevant text as a quote in each of the responses.

Reviewer #1 (Comments for the Author):

In this paper, Midani et al. characterized the metabolic diversity of 88 *C. difficile* isolates (which are classified into 22 ribotypes). The authors found that *C. diff* is a generalist, capable of growing on a variety of carbon substrates. They showed that *C. diff* displays ribotype & clade-specific metabolic capabilities, and two emerging Ribotypes 023 and 255 have different metabolic phenotypes. The authors claimed that *C. diff* clade 5, the most evolutionary distant clade, displayed the most robust growth on simple sugars, which poses questions about the adaptation of *C. diff* to the human gut.

Overall, the authors did a good job in the Results section in explaining their analysis in detail. With the comprehensive metabolic characterization of a large number of *C. diff* isolates, this paper is a good contribution to the field. However, the authors need to tone down some of the claims made in the paper, especially on the implications of the study. Although the paper showed that *C. diff* ribotypes have distinct metabolic capabilities that could help them survive in diverse environmental conditions, the phenotype assessed in this study is simply monoculture growth on different single carbon sources, which is not physiologically relevant. It is difficult to draw any correlations on the adaptation of *C. diff* in the human gut (as claimed in the importance section). The authors did not analyze differences in toxin production which is a major determinant of *C. diff* infection, interactions with human gut microbiota, nor consider other *C. diff* life cycles (sporulation & germination). Discussion about this would be good to acknowledge the limitations and scope of the study. Finally, the author's claim regarding *C. diff* clade 5 exhibiting the most robust growth on simple sugars is not supported by the current data. Specific comments are listed below:

We thank the reviewer for their constructive feedback on our work. In addition to our general response to the editor (above), we would like to address this particular comment by the reviewer: "It is difficult to draw any correlations on the adaptation of *C. diff* in the human gut (as claimed in the importance section). The authors did not analyze differences in toxin production which is a major determinant of *C. diff* infection, interactions with human gut microbiota, nor consider other *C. diff* life cycles (sporulation & germination)."

Our work provided clear evidence about which carbon substrates *C. difficile* can consume and how this consumption varies between ribotypes and phylogenetic clades. Therefore, we revised our manuscript to focus on this strength of the manuscript and therefore do not address additional colonization factors (beyond growth on carbon sources) relevant to the human gut such as toxins or sporulation. We address this limitation of our work in lines 379-382 as follows:

"Finally, in vitro growth of C. difficile on individual carbon sources does not reflect how C. difficile interacts with the complex metabolic diversity available in the human gut, nor does it address other factors that impact C. difficile colonization in the human gut, including sporulation, toxin production, and interaction with other gut microbes."

Regarding this comment by the reviewer: "It is difficult to draw any correlations on the adaptation of *C. diff* in the human gut (as claimed in the importance section)." We revised our manuscript to clarify that some aspects of our study address open questions about the evolutionary adaptation of *C. difficile* to nutrients in the human gut. These questions stem from recent findings (Reference #28) which showed using large-scale genomic analysis that genetically newer Clades 1-4 have experienced positive selection on genes involved in metabolism of simple dietary sugars. Based on this genomic and additional experimental evidence, the authors postulated that evolution of Clades 1-4 of *C. difficile* selected for metabolizing simple sugars. We now provide this information in lines 334-338 as follows:

“A large-scale genomic analysis suggested that the divergence of Clades 1-4 from Clade 5 is linked to the metabolism of dietary simple sugars, for example, by identifying positive selection on genes involved in fructose metabolism, glycolysis, sorbitol, and ribulose metabolism (28). A related mouse experiment also showed that fructose and glucose could differentially impact colonization of C. difficile Clades 1-4 (28).”

Accordingly, we hypothesized that *C. difficile* Clades 1-4 would grow to higher optical density on simple sugars. However, we found that there are no significant differences between Clades 1-4 and Clade 5 on various simple sugars. Outcomes of the mouse experiments in Ref #28 may certainly be explained by factors other than higher growth on simple sugars. So, we provide the following context in lines 342-347 as follows:

“Yet, there are several potential biological explanations for higher colonization of R20291, in contrast to M120, in mice fed simple sugars, including non-regulatory factors, sporulation, toxin production, or sugar-microbiome interactions. So, while our findings highlight the need for more representation of a pathogen’s diversity in animal studies, it remains possible that simple sugars differentially impact the growth of newer C. difficile lineages in the gut.”

Introduction

1. In the introduction, the authors motivate the study by saying that diet could impact gut microbiota composition, modulate *C. diff* disease severity, and contribute to the emergence of epidemic lineages, but how human diet impacts *C. diff* epidemiology and pathogenesis is unclear. However, the paper did not use human diets, but rather simple sugars. Human diets consist of polysaccharides, which cannot be consumed readily by *C. diff*. There are inter-species interactions with the gut microbiota (e.g. fiber degraders with PULs) that mediate *C. diff* growth & colonization.

The dietary and microbial environment in the intestines is complex and, as the reviewer points out, diet-microbial interactions impact *C. difficile* growth and colonization. Yet, we caution about the simplification that “human diets consists of polysaccharides” and the implication that our paper only looked at “simple sugars”. First, the human diets consists of a lot more than polysaccharides; for example, see recent profiling of human intestinal metabolome across the intestinal tract at <https://doi.org/10.1038/s42255-023-00777-z>. Second, we assessed the growth of *C. difficile* on 190 carbon substrates, and only 6 out of the top 26 substrates that we identified (Figure 1) are simple sugars. Nonetheless, we understand the reviewer’s concerns and acknowledge that our introduction was not precise about the aims of our study. So, we revised lines 75-81 to the following:

“Therefore, nutritional and microbial alterations in the gut due to factors other than antibiotics may also increase the risk of C. difficile infections. As a strong driver of both nutritional and microbial composition in the gut (4), the human diet modulates the risk or severity of C. difficile infection in animals (5, 6), and dietary additives may have contributed to the emergence of epidemic C. difficile lineages (7). A deeper understanding of the nutritional preferences of C. difficile can therefore inform dietary and microbial strategies for the prevention and treatment of C. difficile infections.”

In agreement with these changes, we also revised the abstract in lines 21-24 to more clearly define the study’s objectives as follows:

“Although C. difficile is a genetically diverse species, certain ribotypes are overrepresented in human infections and it is unclear if metabolic adaptations are essential for the emergence of these epidemic ribotypes. To identify ribotype-specific metabolic differences, we therefore tested carbon substrate utilization by 88 C. difficile isolates and looked for differences in growth between 22 ribotypes.”

2. As written in the Introduction section, the fact that C. diff is a generalist species capable of growing in multiple substrates is not new. There are a lot of other previous studies that have shown this through growth assays on different substrates and also metabolomics experiments. This is perhaps not a suitable claim to be made in the abstract, or at least the authors should revise the claim that their study "supports the idea that C. diff is a generalist".

We did not intend to claim that the generalist nature of *C. difficile* is an original finding in our study, so we agree with the author. We revised the problematic statement in lines 24-25 of the abstract as follows:

“As expected, C. difficile was capable of growing on a variety of carbon substrates.”

3. The author's claim in the introduction "it is unknown if clinical C. diff isolates have similar metabolic capabilities as lab-adapted strains." is not true. A recent study (<https://doi.org/10.1038/s41467-024-51062-w>) investigated the metabolic capabilities of 19 C. diff isolates and also a lab strain C. diff DSM 27147 (R20291) to study differences in inter-species interactions, which should be acknowledged. In this paper, the authors performed a more comprehensive metabolic assessment with a larger sample size (88 isolates) and focused on whether certain ribotypes have metabolic advantages over others, which should be the point they emphasize.

We thank the reviewer for highlighting the strength of our study as a more comprehensive metabolic assessment with focus on differences between ribotypes. Although our study profiled both lab-adapted and clinical isolates, the final version of our manuscript did not explicitly discuss differences between clinical and lab-adapted strains. So, we revised the introduction now to only focus on our primary study's objective of identifying differences between ribotypes (not clinical versus lab-adapted). Accordingly, we revised lines 89-94 to the following:

“Metabolic capabilities including fermentation of amino acids and central carbon metabolites also varied considerably between strains comprising the five major clades (16). Yet, how these capabilities vary between different ribotypes remains incompletely understood. Broader studies of C. difficile metabolism are therefore needed to identify ribotype-specific metabolic capabilities. By discovering metabolic differences between dominant, emerging, and rare ribotypes, these studies can also improve our understanding of C. difficile evolution and epidemiology.”

In addition, we thank the reviewer for bringing this recent study to our attention (now included in the manuscript as reference #34 in response to reviewer's comment #3 for the Results section). Yet, we disagree that this recent study reveals differences between clinical isolates and lab-adapted strains. It only reveals how a specific lab-adapted strain DSM 27147 (R20291) is different from the 18 clinical isolates tested. Because the typing information of these clinical studies are not included in the study, it is not possible to assess whether the differences that are detected by this study are simply differences between ribotypes rather than habitats (lab-adapted vs clinical)

4. Lines 61: Referencing ref. 8 two times in the same sentence: "Because it is adept at fermenting both amino acids and carbohydrates (8, 8)".

We removed the duplicate reference.

Results

1. The authors should justify the use of Carrying capacity as a metric to measure *C. diff* growth and mention it in the text. In many laboratory media conditions, *C. diff* growth comprised of logistic growth followed by a highly variable stationary phase with a steep reduction in OD600.

The reviewer is correct that *C. difficile* can undergo a steep reduction in optical density in certain growth conditions. This is precisely why we used carrying capacities instead of areas under the curve. The carrying capacity metric is not impacted by this growth behavior because it is defined as the "total growth" which is the difference between the initial optical density and the maximum optical density reached at any time during growth. This rationale and definition are now further highlighted in lines 128-131 as follows:

"Under certain media conditions, C. difficile cultures experience abrupt declines in optical density during stationary phase (17, 18). So, we estimated the maximum population size for each monoculture using the carrying capacity metric (total growth) which is defined as the difference between the initial and maximum optical density during the growth experiments."

2. Since the minimal media contain amino acids and some of them are major resources for *C. diff* growth, e.g. proline, the plots in Figure 1 are therefore the net change in *C. diff* growth from baseline growth under amino acids, which the Authors acknowledged. This sentence might not be true "*C. difficile* reached the highest biomass on simple sugars, followed by sugar derivatives, sugar alcohols, amino acids, then carboxylic acids" since the authors are comparing growth from a baseline growth under amino acids.

We agree with the reviewer. We revised the text to qualify the relevant statement in lines 144-147 as follows:

"C. difficile reached the highest biomass on simple sugars, followed by sugar derivatives, sugar alcohols, amino acids, then carboxylic acids. However, because baseline amino acids in the minimal media are fermented by C. difficile, normalized carrying capacities for amino acids underestimate their total contribution to C. difficile growth."

3. The fact that some resources (e.g. trehalose) show higher COV in growth between isolates is interesting and should be mentioned in the results. This could be because some ribotypes are better at utilizing trehalose than others (<https://doi.org/10.1073/pnas.2119396119>). This was observed in a previous paper from the authors (<https://doi.org/10.1073/pnas.2119396119>) and another study (<https://doi.org/10.1038/s41467-024-51062-w>), which should be cited. I think this is mentioned in the Discussion a bit in Paragraph 2 but also needs to be mentioned in the Results section. The resources with higher COV are likely the ones that appear in the principal component loadings in Fig. 2e-f that contribute to the different clusters of clades/ribotypes based on growth phenotypes.

We agree with the reviewer that additional discussion of this observation is needed in the manuscript, but we restrict this discussion to the Discussion section, in order to keep the Results section focused on the actual experiments and observations detected. Accordingly, we revised the text as follows.

First, we highlight the relationship between the coefficients of variation and the substrates (including trehalose) which describe most of the variation in the principal component ordination in lines 173-174 as follows:

“As expected, normalized carrying capacities on these substrates were highly variable between isolates (see coefficients of variation in Figure 1).”

Second, we explicitly discuss the implications of the coefficients of variation for trehalose and other substrates in the Discussion in lines 318-320 as follows:

“These substrates include trehalose, sorbitol, and ribose, which were associated with high coefficients of variation in our Biolog assay and explained significant variation in the clustering of isolates by their carbon substrate utilization.”

Finally, we cite additional work that displays clear variation in trehalose metabolism by *C. difficile* isolates including the references suggested by the reviewer in lines 320-322 as follows:

“We and others have previously shown that trehalose metabolism is variable between ribotypes and that increased use of trehalose in food manufacturing correlated with the emergence of ribotypes that are highly adept at consuming trehalose (7, 29, 33, 34).”

4. Provide exact p values in Fig. 4b. Although the trend is there, it is unclear if the difference between RT255 & others is statistically significant from the plots.

To address the reviewer’s question, we simplified the analysis of Figure 4b. Instead of linear models, we directly test the hypothesis that Ribotype 255 isolates *grow to higher* carrying capacity under the tested conditions than Other isolates. So, we used one-sided Wilcoxon rank-sum tests, corrected for multiple comparisons, and added corresponding asterisks directly in Figure 4. We refer to these changes in the Figure caption in lines 740-741 as follows:

*“*P < 0.05, **P < 0.01, ***P < 0.001, one-sided Wilcoxon rank-sum t-test with false discovery rate correction. NS indicates not significant tests.”*

We also modified the text accordingly in lines 217-219 as follows:

“As expected, we observed significantly higher growth for ribotype 255 on fructose and ribose (P < 0.05, one-sided Wilcoxon rank-sum tests with false discovery rate correction; Figure 4B)”

5. Lines 147-148 & Figure 4b: Why fructose and ribose specifically? The authors should explain the rationale why they pick these two, since there are many other substrates where RT255 has higher growth (Fig. 3).

Per the reviewer’s suggestion, we explain the rationale in lines 210-215 as follows:

“We observed that ribotype 255 was significantly enriched on various substrates (Figure 3). These substrates included fructose which is a common sugar in the large intestine because of

high dietary intake in the Western world (24, 25), and ribose which had the second highest coefficient of variation in the Biolog assays (Figure 1). To verify these observations, we re-tested the growth of ribotype 255 and other isolates representing various common ribotypes on minimal media supplemented with fructose or ribose at multiple concentrations.”

6. I am not convinced by the claim that "Clade 5 ribotypes grow more robustly on simple sugars than Clade 1-4 ribotypes". From Fig. 3, we see that Clade 5 has a negative enrichment score in many substrates. The experiments in Fig.5 were merely based on 8 substrates (with 4 simple sugars: glucose, fructose, tagatose, and ribose). The results show that Clade 1-4 and Clade 5 have similar growth overall ($P>0.05$) and Clade 5 grew to a higher density only on two sugars (ribose and tagatose). This does not support the author's claim "Clade 5 ribotypes grew more robustly on simple sugars than common ribotypes in Clades 1-4". The authors may need to do more validation experiments with more simple sugars to be able to draw a clear conclusion.

To address the reviewer concern's, we applied a more conservative statistical test, instead of linear models, by simply using Student's t-test and adjusting the p-values using the Benjamni-Hochberg method. This approach did not detect any significant differences which we report in lines 254-261 as follows:

“Based on the positive selection on genes involved on metabolism of simple sugars (28), we hypothesized that Clades 1-4 would reach higher total growth on simple sugars. However, we did not detect any statistically significant differences between Clade 5 and other isolates on the eight tested substrates ($P>0.05$, Student's t-test with false discovery rate correction). On the contrary, we detected statistical trends for higher growth of Clade 5 isolates on two simple sugars, ribose and tagatose, but additional experiments are needed to confirm these differences. In summary, Clade 5 ribotypes grew robustly on simple sugars to a comparable level as Clades 1-4 ribotypes.”

We also revised the section title in line 226 as follows:

“Clades 1-4 and Clade 5 ribotypes display similar growth on simple sugars”

and the Figure caption in line 743 as follows:

“Figure 5. Clade 5 ribotypes grow robustly on simple sugars similar to Clade 1-4 ribotypes.”

We also added the following clarification to Figure 5 caption in lines 751-753:

“We did not detect any significant differences between Clade 5 and Other clades on each of the tested substrates ($P > 0.05$, Student's t-test with false discovery rate correction).”

We revised our abstract in lines 28-29 as follows:

*“In addition, although *C. difficile* Clade 5 is the most evolutionary distant clade and often detected in animals, it displayed robust growth on simple sugars similar to Clades 1-4.”*

and the Importance section in lines 54-56 as follows:

“We also found that genetically newer and older clades grew to a similar level on simple sugars, which contrasts recent findings that newer clades experienced positive selection on genes involved in simple sugar metabolism.”

7. "Altogether, this precarious growth of M68 suggests that an unknown limiting factor can modify the growth of some *C. difficile* strains on certain substrates, including trehalose, ribose, and cellobiose." Where is the validation data for ribose and cellobiose? The authors only provide validation experiments for trehalose in Fig. S5"

We only performed the validation for trehalose, so we clarified lines 280-282 as follows:

"Altogether, this precarious growth of M68 suggests that an unknown limiting factor can modify the growth of some C. difficile strains on trehalose."

In addition, we included a statement in the discussion to clarify that additional studies are needed in lines 363-365 as follows:

"Because we only validated that this limiting factor supports growth of M68 on trehalose, additional studies are needed to verify if growth factors in yeast extract also facilitate higher growth of other strains on trehalose and higher growth of M68 on other substrates including cellobiose, ribose, and xylose."

Discussion

1. Paragraph 2: It is good that the authors discuss how the sugars they used in this study could be relevant in human diets, but the authors should also acknowledge/discuss that these simple sugars may not be physiologically relevant, as human diets consist of polysaccharides (e.g. dietary fibers) which are not explored in this study.

We disagree with the reviewer's simplification that "human diets consist of polysaccharides". Numerous studies have measured the metabolites in the intestinal lumen and feces of both humans and animals and found a rich diversity of nutrients that are accessible by gut microbes including *C. difficile* (for examples, see <https://doi.org/10.1038/s42255-023-00777-z>). Further, through cross feeding, polysaccharide degradation by gut bacteria can provide simpler nutrients, including the monomers comprising these polysaccharides, into the lumen where other microbes can consume it (for example, see <https://doi.org/10.1038/s41467-021-27191-x>, <https://doi.org/10.1080/19490976.2021.1993582>, and <https://doi.org/10.1128/AEM.72.5.3593-3599.2006>). We also disagree that "simple sugars may not be physiologically relevant". As we discuss in this paragraph, trehalose, fructose, and sorbitol are highly prevalent in the Western diet (see References #25 and #36) and frequently detected in human and animal intestines/stool (see References #11, #24, #35, and #38) especially during inflammation and after antibiotic exposure which is the primary risk factor for *C. difficile* infections.

Finally, we do not explore polysaccharides because our study focuses on substrates that *C. difficile* can consume and there is no clear evidence in the field whether *C. difficile* can consume polysaccharides. Instead, we focused on substrates that can be profiled using commercially available Biolog PM plates (a total of 190 substrates), and we acknowledge the limitation of these 190 substrates in lines 368-369 as follows:

"Our analysis was limited to the 190 carbon substrates included in the Biolog Phenotype Microarray plates"

2. The authors already discussed some of the limitations and the scope of the study in the second last paragraph. It will be good to incorporate other limitations mentioned above in this paragraph.

See response to the previous comment.

Reviewer #2 (Comments for the Author):

In this manuscript, Midani and colleagues profile carbon substrate utilization of a collection of *C. difficile* clinical isolates to establish links between carbon metabolism and evolution or emergence of *C. difficile* ribotypes. The authors report that *C. difficile* clinical isolates are generally capable of growth on many carbon sources. Carbon source preference clustered by both clade and ribotype, indicating phylogenetic conservation of metabolic capabilities. Importantly, they found that two emerging *C. difficile* ribotypes displayed differences in preferred carbon sources, suggesting that carbon metabolism is not a main driver of ribotype emergence. Overall, there is high enthusiasm for this work. These data provide clinical relevance to metabolic studies performed in laboratory strains of *C. difficile* and provide new insights into important clinically relevant strains of this important pathogen. Some minor comments below:

We thank the reviewer for their enthusiasm and thoughtful comments.

1. The description of Figure 4 in the text is a bit confusing. Figure 4A is referenced, but not described in detail. In addition, the text does not reflect the fact that the metabolic phenotypes of RT023 were also experimentally confirmed, and instead the text focuses entirely on RT255. Also added context for importance of utilization of fructose and ribose by RT255 is needed.

Regarding the comment: “The description of Figure 4 in the text is a bit confusing. Figure 4A is referenced, but not described in detail.” To clarify the purpose and content of Figure 4A, we revised the text in lines 195-202 as follows:

“By focusing only on significant enrichments, we surprisingly observed that ribotypes 255 was the most positively enriched ribotype while ribotype 023 was the most negatively enriched (Figures 3 and S4A), with a total of 14 substrates that were significantly associated with either or both of these emerging ribotypes (Figure 4A). Comparison of the total growth of these ribotypes on these 14 substrates displayed a polarized enrichment pattern: Ribotype 255 and 023 displayed very high or very low growth, respectively, in comparison to other isolates in the study (Figure 4A).”

We also revised the caption for Figure 4A in lines 721-726 as follows:

“(A) Plots compare the normalized carrying capacity for ribotype 255 and 023 isolates to other isolates on the 14 substrates that were significantly associated with these two emerging ribotypes using strain set enrichment analysis (see RT255 and RT023 columns in Figure 3). Substrates that were significantly associated with both ribotypes are labeled with black text, whereas substrates that were significantly associated with only ribotype 255 or ribotype 023 were labeled with blue and green text respectively.”

Finally, we added the following in lines 727-728 to clarify that Figure 4B include additional validation beyond what is included in Figure 4A:

“(B) In additional validation experiments, “RT255” isolates reached significantly higher carrying capacity on fructose and ribose than isolates belonging to “other” ribotypes.”

Regarding the comment: “In addition, the text does not reflect the fact that the metabolic phenotypes of RT023 were also experimentally confirmed, and instead the text focuses entirely on RT255”. The reviewer is correct that we mostly discussed validation for growth of RT255 but not RT023 isolates. As shown above, our revisions in lines 195-202 and 721-726 should clarify that we contrasted metabolic phenotypes of RT023 and RT255 based on the Biolog assay (Figure 4A) but then provide additional validation only for R255 (Figure 4B). We focused on additional validation for RT255 growth because it is reasonable to assume that the heightened growth of RT255 on carbon substrate may provide a competitive advantage in colonizing the human gut, especially in contrast to other *C. difficile* ribotypes. Conversely, our Biolog results indicate that RT023 overall grows poorly on most substrates (relative to other *C. difficile* isolates) and we only detected significant negative enrichments for RT023 (Figure 3). So, we speculate that carbon metabolism is less likely to contribute a fitness advantage to RT023 that would potentially explain its recent emergence.

Regarding the comment: “Also added context for importance of utilization of fructose and ribose by RT255 is needed.” We added our rationale for selecting fructose and ribose for the validation experiments in lines 210-213 as follows:

“We observed that ribotype 255 was significantly enriched on various substrates (Figure 3). These substrates included fructose which is a common sugar in the large intestine because of high dietary intake in the Western world (24, 25), and ribose which had the second highest coefficient of variation in the Biolog assays (Figure 1).”

2. The transition to Figure 6 seemed abrupt and more rationale is needed. Namely, the introduction to trehalose utilization would be useful. For instance, what is the effect of the mentioned C171S amino acid substitution? Not clear without exploring previous literature that this mutation enhances utilization, not decreases.

We agree with the reviewer that the transition into this section and Figure 6 was inadequate. So, we revised lines 265-272 as follows to highlight the rationale.

“Thus far, we validated several aspects of our broad survey of carbon substrate utilization, including the heightened growth of Ribotype 255, and the comparable growth of Clades 1-4 and Clade 5 on various simple sugars. However, our survey also identified growth patterns that did not completely agree with prior work. In particular, we previously showed that ribotypes 027, 078, and 017 encode genetic variants or operons that facilitate competitive growth on trehalose (7, 29). Here, using our Biology assay, we confirmed that ribotypes 027 and 078 indeed grow robustly on trehalose but observed that ribotypes 017 isolates do not. So, we re-tested M68, a ribotype 017 reference strain, on Biolog Phenotype Microarray plates using different growth media for its overnight culture, before dilution and inoculation into Biolog plates.”

In regard to the C171S amino acid substitution, our previous work (Reference #29) showed that Ribotype 017 isolates can grow on 10 mM trehalose (similar to Ribotype 027 isolates). We also noted that Ribotype 017 have a conserved C171S amino acid substitution in the trehalose operon repressor *treR* (different than the RT027 *treR* variant). We suspected that his substitution precisely enhances growth on trehalose because it is near the predicted effector binding pocket of TreR. However, because

the amino acid substitution is not central to the rationale of this section, we no longer discuss it in this manuscript.

3. Experiments highlighted in Figure 6 and accompanying conclusions need added experimentation or description to provide insights into these observations. What factors in yeast extract are driving this phenomenon? Is there alteration in activity or expression of the trehalose utilization operon in high, low, and no yeast extract cultures? Is the trehalose utilization operon required for this observation? Do trehalose adaptation experiments (low level trehalose in overnights followed by growth in high levels of trehalose) provide similar results? It is noted that RT027 does not respond to washing, but there is a seemingly significant decrease in growth following washing.

The reviewer highlights some important questions about the phenotype that we observed in Figures 6 and S5. However, we believe that interrogation of this phenotype is beyond the scope of this study. In addition, the reviewer points out that washing decreases growth of RT027 on trehalose. However, RT027 is still able to grow on trehalose even after washing or heavy dilution of inocula. Of course, this indicates that there are *additional* factors in yeast extract that increase the growth of RT027 on trehalose, but these additional factors do not seem to be *limiting* (i.e., RT027 can still grow on trehalose without them). This is indeed a very interesting observation, but it does not contradict our interpretation of the results. This observation should be also further investigated, but this would be beyond the scope of our manuscript as well. In the response to the previous comment, we improved the transition to this section to clarify the scope of this section in the context of our manuscript.

4. The authors note in their methods that they calculate area under the curve and growth rates, but these data are not included. In addition to carrying capacity, understanding how different strains growth dynamics differ under different conditions would be quite informative. Presumably ranking of substrates could be done with growth rate, as well? Was this analysis performed? If not, this should be noted as a limitation in the discussion, as carrying capacity isn't necessarily the most important factor in fitness/virulence.

We previously performed the analysis using growth rates (similar to Figures 1 and 2) but did not include them in the manuscript, because they did not reveal any significant additional insight. However, as the reviewer points out, these results may still be useful for readers, so we now include four additional supplemental figures (Figures S1, S2, S4B, and S5).

We highlight the results from Figure S1 in lines 150-156 as follows:

“For growth on the top 26 substrates, we also estimated growth rates for all isolates (Figure S1) and found that medians of normalized carrying capacities were highly correlated with medians of normalized growth rates (Pearson’s correlation coefficient = 0.87, $P < 0.001$). However, there were slight deviations from this correlation: beta-hydroxybutyric acid and the amino acids leucine, threonine, and L-hydroxyproline had higher normalized growth rates than expected based on their normalized carrying capacities, while N-Acetyl-Neuraminic acid, mannose, arbutin, salicin, arabitol, and sorbitol had lower normalized growth rates than expected.”

We also highlight the results from Figure S2 in lines 168-169 as follows:

“A similar principal component analysis but using the normalized growth rates for the top 26 carbon substrates detected similar but weaker clustering patterns (Figure S2).”

We also highlight the results from Figure S4B and S5 in lines 202-203 as follows:

“Furthermore, we performed a similar enrichment analysis but using normalized growth rates and we detected similar patterns but fewer significant enrichments overall (Figure S4B, S5).”

Re: mSystems01075-24R1 (Emerging *Clostridioides difficile* ribotypes have divergent metabolic phenotypes)

Dear Dr. Firas S Midani:

Both reviewers have now evaluated your revised manuscript and are content with the revisions. Therefore, I am delighted to accept your manuscript for publication. However, I would ask you to please review the statement/phrase mentioned by the first reviewer.

Your manuscript has been accepted, and I am forwarding it to the ASM production staff for publication. Your paper will first be checked to make sure all elements meet the technical requirements. ASM staff will contact you if anything needs to be revised before copyediting and production can begin. Otherwise, you will be notified when your proofs are ready to be viewed.

Sincerely,
Christian Diener
Editor
mSystems

Reviewer #1 (Comments for the Author):

The authors have done a good job in revising the paper. I believe the paper is now suitable for publication in mSystems. I just have one more minor comment:

Could the author re-check this sentence in Lines 214-216: "On the contrary, we detected statistical trends for higher growth of Clade 5 isolates on two simple sugars, ribose and tagatose, but additional experiments are needed to confirm these differences."? To my understanding, the new statistical test indicates there's no statistical difference, which is shown by NS in Figure 5? This also contradicts with the previous sentence " However, we did not detect any statistically significant differences between Clade 5 and other isolates on the eight tested substrates ($P > 0.05$, Student's t-test with false discovery rate correction)."

Reviewer #2 (Comments for the Author):

In this revised manuscript, Midani and colleagues have provided an in-depth responses to my concerns, as well as the other Reviewer. The manuscript is markedly improved through clarification to text and methods, and new experimentation and analyses. Overall, I am enthusiasm about this work and the new insights into clinically relevant strains of this important pathogen will be of great interest to the field.